# Investigation of a Base-Isolator System's Effects on the Seismic Behavior of a Historical Structure

**Pınar Usta**

Department of Civil Engineering, Isparta University of Applied Science, Isparta 32200, Turkey; pinarusta@isparta.edu.tr; Tel.: +90-246-214-6812

**Abstract:** The earthquake performance of structures with seismic isolation is much better than that of fixed-base structures, and the application of seismic insulation ensures both structural integrity and the protection of the items present in the structures. The base-isolation system is used to extend the fundamental period of vibration of the structure and to obtain higher value from base-isolated structures relative to the fixed-base structure. Historical masonry mosques could be strengthened using a base-isolation technique. In this study, a historical masonry mosque was organized and modelled using SAP2000 software. Nonlinear Time History analyses were carried out for the historical masonry structure, firstly for the fixed-base mosque and secondly for the base-isolated mosque with lead rubber bearing (LRB). The use of a base-isolator system caused an increase in the historical mosque's period, reducing the displacements, acceleration, and force applied on the mosque and the resulting structural deformation; the results of the analysis indicate a significant improvement in the seismic behavior. The modelling results show that such historical masonry buildings (especially those with high and delicate minarets) can be vulnerable to major earthquakes, and it may be useful to examine strengthening strategies for these buildings.

**Keywords:** historical masonry structure; seismic isolation retrofit; seismic vulnerability assessment





## 1. Introduction

Historical buildings are the permanent memory of urban development [1]. The conservation of heritage buildings and their association with appropriate structural protection principles is a historical, cultural, and engineering process that requires a multidisciplinary and multicultural approach [2–4]. Preserving the heritage built in developed societies and transferring it to future generations is very important to consolidate a collective memory that creates a sense of belonging for citizens and access to cultural heritage [5–7]. Efforts are needed to ensure the sustainable improvement and preservation of cultural resources for future generations [8]. Therefore, it is important to investigate the restoration of historical buildings compared to non-historic buildings to improve sustainable recovery [2,9,10].

In 19th-century Ottoman mosques, the plan type of a single-domed square space is the basic architectural scheme, and the number of these mosques is quite high. Many of these surviving mosques have been the subject of various publications [11]. Considering their historical features and cultural significance, the examination of single-domed mosques and their strengthening when necessary are very important in terms of their survival of possible earthquake effects.

In Turkey, which is defined as an earthquake zone, whether a structure can withstand seismic effects is an important question. In particular, the performance of buildings in old building areas is important for earthquake engineering [12]. Earthquakes are natural disasters that cause loss of life and considerable economic damage [13–15]. The classical design approach to designing buildings that are safe against earthquakes is increasing the ductility of structures. The installation of seismic isolation systems in structures is an effective design approach for the reduction of earthquake damage [16–18]. Seismic base-isolation technology aims to reduce the seismic forces acting on the building by extending

the duration of the building's natural period instead of increasing the earthquake resistance capacity of the structure. Extending the natural period of the structure from the prevailing frequency of ground movements is based on the principle of significantly reducing the acceleration transmitted to the superstructure. The separation of the superstructure and the foundation via sliding and flexible systems placed on the base forms the basis of seismic isolation [17–23]. This application gives the building a base period much higher than the fixed-base duration. The period shift caused by the insulation system causes a decrease in the acceleration, and this decreases the inertial force value affecting the structure [24–26]. The isolation system absorbs some of the earthquake energy and increases the period by transferring the remaining structure to the superstructure. Displacement limits at the isolation level are determined by damping or energy-emitting capacity isolators [26,27]. The base-isolation system is one technique used to reduce earthquake hazards in historic masonry buildings and is a method of dissipating energy rather than a structural improvement. Thus, the earthquake resistance of the building increases. Isolators aim to reduce the destructive effects of earthquakes by giving the structure a period greater than the earthquake prevailing period [17,28]. The base-isolation system is thus an attractive retrofit option for historical buildings or essential facilities. In this system, alterations to the superstructure are significantly reduced or removed, and the basic vibration period of the building is shifted to a range other than the dominant energy content of the earthquake [29,30].

In Turkey, which faces earthquakes frequently, the strengthening of historic buildings with base isolators is still not common. The repair and protection of historical buildings are still provided by way of interventions to the superstructure. However, this situation alters historical buildings from their original forms, and it creates results that change the structure and are not suitable. There are several studies in the literature on the seismic performance of historical masonry structures. Some of them are listed below. Soyluk and Tuna (2011) performed dynamic analysis of the historical Sehzade Mehmet mosque for base-isolation applications [31]. Clemente and De Stefano (2011) studied the application of seismic isolation in the retrofitting of historical buildings [32]. Sezen (2012) researched the seismic vulnerability and preservation of historical masonry monumental structures [33]. Erkek et al. (2013) studied the seismic behavior of the historical Malatya Grand Mosque. Usta and Bozdağ (2020) researched the structural performance of Izmir Basdurak Mosque [34]. Usta and Bozdag (2019) carried out an investigation of the earthquake behavior of Worship Buildings [35]. Aras et al. (2019) researched seismic isolation retrofitting of a historical masonry structure [36,37]. Yazgan et al. (2019) examined the effects of interventions and additions on the structural performance of the Imaret Section of Sinan Pasa Kulliye [38]. Kıpcak et al. (2017) carried out an investigation on the effects of seismic isolation on the structural performance of Zalpaşa mosque as a function of ground motion magnitude [10]. Lagomarsino and Podesta (2004) carried out damage and vulnerability assessments of churches after the 2002 Molise, Italy, earthquake [39]. Matteis et al. (2019) investigated a predictive methodology for vulnerability assessment of churches in large territorial areas [40]. Ruggieri et al. (2020) described the seismic vulnerability of a sample of 90 masonry one-nave churches, subjected to the Valle Scrivia Earthquake, 2003 and supported similar studies describing masonry buildings [41,42]. Liu et al. (2021) investigated the earthquake damage to various engineering structures in Nepal [43]. Liu et al. (2021) analyzed the nonlinear response of an isolated structure under a near-fault earthquake to evaluate the performance of base-isolated structures under near-fault earthquakes [44].

In this study, it is strongly emphasized that buildings with high historical value can be easily strengthened and protected against earthquake effects with the help of isolators without damaging the structure. The results of this study emphasize the importance of using insulators in the protection of historical buildings in Turkey in the future.

The present paper is organized into eight sections. The Section 2 gives details about the historical mosque and clarifies the intended purpose of the article. The Section 3 provides

earthquake data. The Section 4 deeply explains the isolator. The Sections 5 and 6 explain the difference between both systems. Finally, in the Section 7, the study results are briefly evaluated, and improvements that could be made to protect historical buildings in further studies are suggested.

## 2. Building Description and Numerical Modelling

### 2.1. The Ulu Mosque

The studied historical masonry mosque is located in Afyon, Turkey. This historical mosque withstood the Sandikli Earthquake of magnitude 9 in 1875. In this earthquake, some cracks were formed in the walls of the Ulu mosque, but many houses in the region were destroyed. The Grand Mosque has a special structure unlike any other mosque in our country. There are no columns in its structure. The dome structure is also different from that of other historical mosques. The mosque has a square plan with a single dome. The dimensions of the historical mosque are 17.1 m × 17.1 m in length and breadth, 16.84 m in dome height, and 30.59 m in minaret height. The walls of the masonry mosque are built with regional stones and bricks, although the minaret is constructed with only stone. The main dome was built from masonry bricks. The wall thickness is nearly 1.4 m. The historical Ulu Mosque and its location in Turkey are shown in Figure 1.

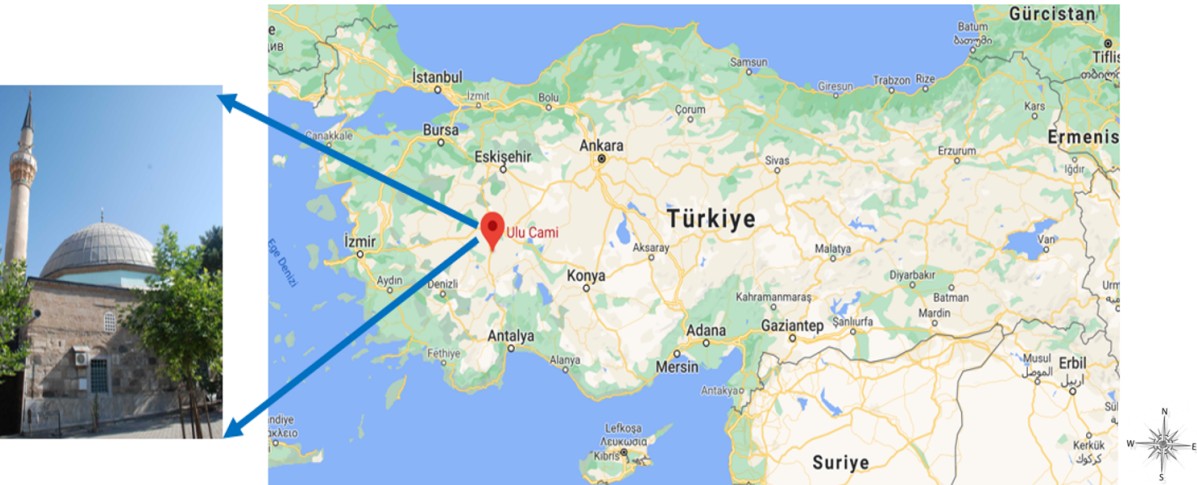

**Figure 1.** Ulu Mosque and its location in Turkey.

### 2.2. Experimental Tests to Define of the Properties of Afyon Tuffs Stone

The Afyon volcanic stone, which has an important place in Inner West Anatolia, covers wide open areas in the region between Bayat, İscehisar, Kırka, Sandikli, and Suhut. During the Ottoman period, various historical buildings such as mosques and fountains in Afyonkarahisar were made of tuffs. Ayazini tuffs have been traditionally used as a building material in many areas in local building constructions in the region since pre-historical times. Despite the fact that tuffs have relatively low durability and low strength values compared to marble, etc., they have survived with no major deterioration failures on many historical buildings [45].

Celik et al. (2016) performed laboratory tests on the tuffs according to TS and ASTM standards [46]. The mechanical properties of the Afyon tuffs rocks are given in Table 1.

**Table 1.** The mechanical properties of Afyon tuffs rocks.

|  | Min | Max | Av. |
|---|---|---|---|
| Unit Volume Weight (kN m$^3$) | 1.40 | 1.50 | 1.45 |
| Density | 2.43 | 2.49 | 2.45 |
| Ultrasonic Wave Speed (km s$^{-1}$) | 2.017 | 2.517 | 2.32 |
| Compressive Strength (MPa) | 19.56 | 24.19 | 21.22 |
| Compressive Strength After Freezing (MPa) | 15.99 | 22.78 | 18.73 |
| Bending Strength (MPa) | 2.35 | 3.13 | 2.69 |

### 2.3. Material Properties and Numerical Modelling

The structural behavior of the mosque was analyzed using the finite element (FE) technique and a macro modeling strategy. The analysis models were developed using SAP2000 V21 software [30]. The mosque walls and minaret were modeled using solid elements; shell elements were used to model the main dome and cone. The openings in the building were made the same, and nonlinear analyses were performed assuming a rigid ground foundation (fixed-base model). The final 3D model consisted of 9197 points, 732 areas, and 4922 solid elements. The finite element model of the mosque is shown in Figure 2.

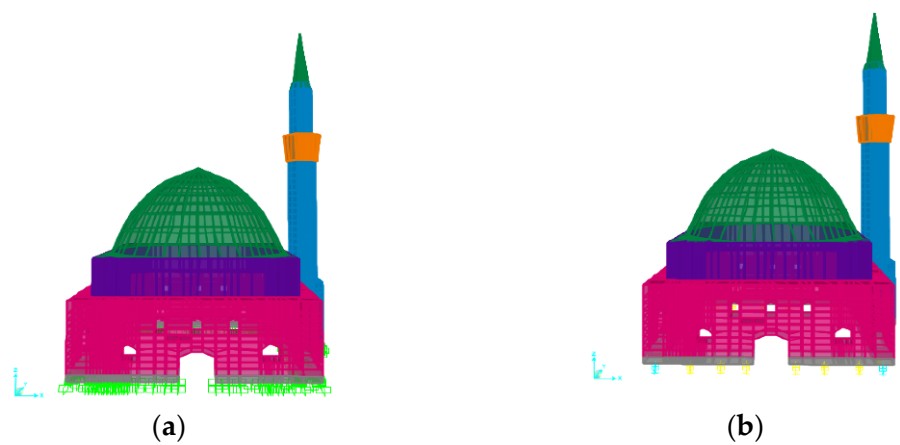

**Figure 2.** Finite element model of the mosque. (**a**) Fixed Base. (**b**) Base Isolated.

The outer walls of the historical mosque with tabhanes were built via a traditional mortar masonry technique. The wall braid used mostly cut stone and some rough stone and bricks. It can be seen that these materials and construction techniques used in the construction of the buildings are related to the political and socio-economic situation and the geographical features of the region where the building is located. It was assumed that alternating stone masonry and brick bond-building elements show a single material feature together with a mortar, and the materials used in numerical modeling have linear mechanical and physical properties. In the model, a rigid slab was placed on the floor of the mosque and 24 seismic isolators identified with link element elements were placed under this slab. The material properties that were used in the finite element model of the masonry structure are shown in Table 2.

**Table 2.** Material properties used in the model (Celik and Gaye, 2016; Aslan and Aslan, 2016).

| Structural Element | Element Type | Model Type | Modulus of Elasticity (MPa) | Density (kg/m$^3$) | PoissonRatio |
|---|---|---|---|---|---|
| Wall | Stone | Solid | 7427 | 2450 | 0.2 |
| Minaret | Stone | Solid | 7427 | 2450 | 0.2 |
| Dome | Brick | Shell | 3000 | 1800 | 0.18 |

## 3. Dynamic Analysis of the Mosque

In this paper, the behavior of the fixed-base and base-isolation cases of the masonry mosque, under earthquake loading, was investigated. For this, nonlinear analyses were performed for the historical mosque. The elastic design spectrum belonging to the place where the mosque is located was determined according to the Turkish Earthquake Code, TBDY 2019 [47]. In the analyses of the mosque, earthquake level DD2, for which the probability of exceedance in 50 years is defined as 10%, was taken as the earthquake level.

Earthquake acceleration records used for analysis in the time domain were taken from the Pacific Earthquake Engineering Research Center (PEER) website [48]. Earthquake analyses were made using 11 earthquake records. In the analyses, a total of three acceleration components, two horizontal and one vertical component, were used for each earthquake. When selecting the earthquake record, the aim was for the average spectra of the earthquake records to be close to the design acceleration spectrum determined for the DD2 earthquake level. The earthquake records were scaled to ensure that the spectra provided the desired properties. The names and characteristics of the earthquake records used in the analysis are given in Table 3. The elastic design acceleration spectrum of earthquake level DD2, the spectra of each earthquake, and the average of these spectra are shown graphically in Figure 3.

**Table 3.** Earthquake records used in the analysis (PEER, 2021).

| ID | Earthquake Name | Year | Station Name | Mag. | Mechanism | Distance (km) |
|----|----------------|------|--------------|------|-----------|---------------|
| 1 | "Parkfield" | 1966 | "Cholame-Shandon Array #12" | 6.2 | Strike slip | 17.64 |
| 2 | "Imperial Valley-06" | 1979 | "Cerro Prieto" | 6.5 | Strike slip | 15.19 |
| 3 | "Imperial Valley-06" | 1979 | "Niland Fire Station" | 6.5 | Strike slip | 35.64 |
| 4 | "Imperial Valley-06" | 1979 | "Parachute Test Site" | 6.5 | Strike slip | 12.69 |
| 5 | "Irpinia_ Italy-01" | 1980 | "Rionero In Vulture" | 6.9 | Normal | 27.49 |
| 6 | "Landers" | 1992 | "Twentynine Palms" | 7.3 | Strike slip | 41.43 |
| 7 | "Kobe_ Japan" | 1995 | "Tadoka" | 6.9 | Strike slip | 31.69 |
| 8 | "Duzce_ Turkey" | 1999 | "Lamont 531" | 7.1 | Strike slip | 8.03 |
| 9 | "Landers" | 1992 | "Forest Falls Post Office" | 7.3 | Strike slip | 45.34 |
| 10 | "Bam_ Iran" | 2003 | "Mohammad Abad-e-Madkoon" | 6.6 | Strike slip | 46.20 |
| 11 | "Darfield New Zealand" | 2010 | "OXZ" | 7.0 | Strike slip | 30.63 |

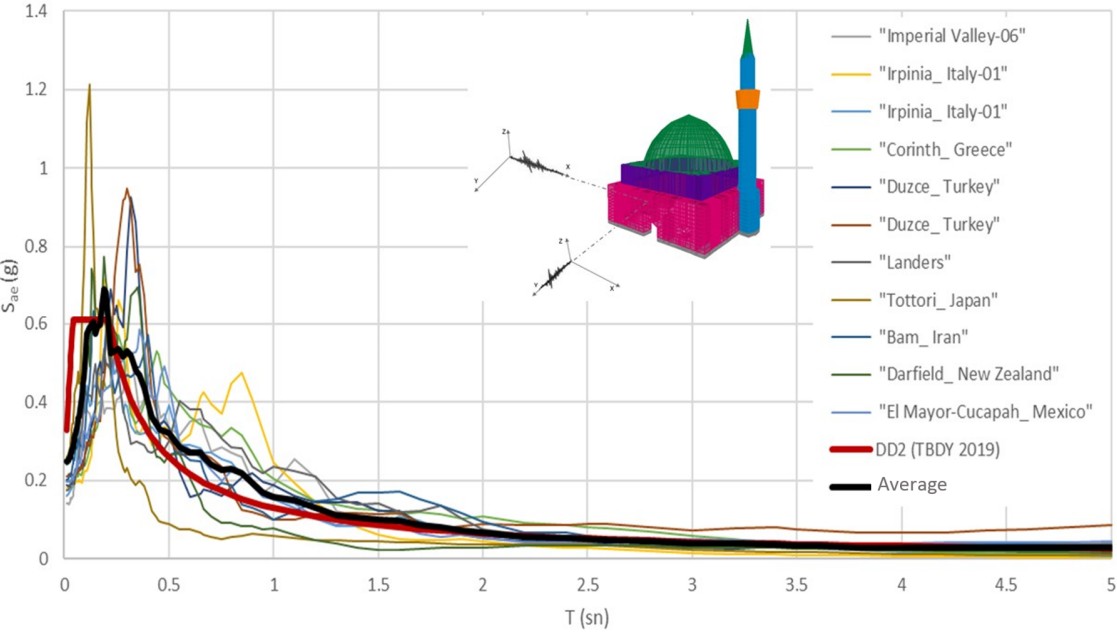

**Figure 3.** The elastic design acceleration spectrum of earthquake level DD2 used in the earthquake analysis for the historical mosque, the spectra of each earthquake, and the average of these spectra.

## 4. Selection and Modelling of the Low Rubber Bearing

In historical and monumental buildings, which are of relatively low height and often massive, the natural vibration periods are rather short. Moreover, their seismic response is primarily determined by brittle failure mechanisms [49,50]. Old and traditionally built structures are generally more strongly affected by earthquakes, but the use of a base-isolation system can significantly increase the earthquake performance of such structures [49,51]. Old buildings are mainly nonreinforced masonry structures that were built mainly based on the experience of masons and builders, without any structural seismic design. In most of these structures, depending on the age of the building, problems related to structural integrity, binding material, connection, load transfer, aging, and the foundation have occurred, and structural deficiencies have occurred due to these problems.

This situation leaves historical buildings vulnerable to seismic activities and makes them much more susceptible to earthquakes than modern buildings [36,52].

It is therefore beneficial to improve the seismic performance of traditional historic structures, particularly those located in seismically active areas. Since it is the most important principle to keep interventions to a minimum in the repair and strengthening of historical buildings, the use of a base isolator in reinforcement becomes an obvious choice as it will help to protect the historical and architectural features of the building [51]. A list of several large retrofit building projects completed using base isolation is given in Table 4 for illustration purposes.

**Table 4.** Major building retrofit projects using base isolation (Matsagar et al., 2008).

| Sr. No | Project and Country | Year | Isolation Systems Utilized |
|--------|---------------------|------|----------------------------|
| 1 | Campbell Hall, Monmouth, Oregon, USA | 1993 | Lead rubber isolator and rubber isolator |
| 2 | Oakland City Hall, Oakland, California, USA | 1994 | Lead rubber isolator and rubber isolator |
| 3 | Long Beach V.A. Hospital, Long Beach, California, USA | 1995 | Lead rubber isolator, rubber isolator, and sliding bearing |
| 4 | Martin Luther King, Jr. Civic Center Building, Berkeley, California, USA | 1995 | High-damping rubber bearing and lead rubber bearing |
| 5 | Kerckhoff Hall, UCLA Campus, Westwood Village, California, USA | 1996 | Lead rubber isolator |
| 6 | San Francisco City Hall and Civic Center, San Francisco, California, USA | 1998 | Lead rubber isolator |
| 7 | Public Safety Building—911 Emergency Communications Center, San Francisco, California, USA | 1998 | Lead rubber bearing and sliding system |
| 8 | Head office of Himeji Shinkin Bank, Himeji Credit Bank, Himeji City, Hyogo, Japan | 2000 | Rubber bearings and dampers |
| 9 | Tokyo DIA Building, Tokyo, Japan | 2001 | Rubber bearings and viscous dampers |
| 10 | Shinjuku Station West Entrance Main Building, Tokyo, Japan | 2002 | Rubber bearings |

The most common base-isolation devices used by engineers for many years are lead rubber bearing (LRB) isolators that combine the isolation function and energy distribution in a single compact unit. For this reason, LRBs were considered in this model. LRBs consist of laminated rubber and lead-core steel plates, an upper plate, a Teflon plate, and a lower link plate. An LRB with a Teflon plate attached to the upper surface of the upper plate of the LRB is inserted into the cavity of four steel blocks to limit shear displacement [19,53].

Due to the multilayer laminated steel plates, these insulators (LRBs) have two factors: they are very hard in the vertical direction and soft in the horizontal direction under seismic loads [54]. Therefore, lateral stiffness potentially increases against strong ground movements. Consequently, the main purpose of adding lead is to increase the stiffness and energy dissipation capacity, both at relatively low horizontal strength levels [55]. For this reason, the LRB system can support the structure vertically, provide horizontal flexibility with the restoration force, and provide the necessary hysteretic damping [19,56,57]. Besides

this, they require minimal costs for installation and maintenance as compared to other passive vibration-control devices, have no moving parts, are not affected by time, and are resistant to environmental degradation [58,59].

The operating principle of this system is similar to that of the laminated rubber bearing system, but the systems are different from each other because a cylindrical lead core is placed in the middle of the LRB system to provide additional rigidity to the system [55]. The plastic behavior of the lead core gives this isolator important hysteretic behavior [57]. The parameters of the two linear approximations expressing the hysteretic law of behavior are as follows:

$$D_y = \frac{Q}{(K_1 - K_2)} \tag{1}$$

$D_y$ is the yield displacement, $D$ is the design displacement of the LRB, and $W_d$ is the energy dissipated by the cycle, corresponding to the design displacement to the total area of the hysteresis loop, given in the formula below:

$$W_D = 4Q(D - D_y) \tag{2}$$

$F_y$ is the yield force in a monotonous loading, and $Q$ is the force, corresponding to null during cyclic loading, and also represents the characteristic strength and yield force of the lead bar for the LRB.

$$Q = F_y - K_2 D_y \tag{3}$$

$F_{\max}$ is the maximum shear force corresponding to the design displacement D, and $K_1$ is the elastic stiffness for monotonous loading, also equaling the stiffness of unloading in cyclic loading.

$$K_1 = F_y / D_y \tag{4}$$

$K_2$ is the post elastic stiffness.

$$K_2 = \frac{(F_{max} - F_y)}{(D - D_y)} \tag{5}$$

$K_{eff}$ is the effective stiffness of the LRB, given by the following formula:

$$K_{eff} = K_2 + \frac{Q}{D} \quad D \gtreqqless D_y \tag{6}$$

$B_{eff}$ is the effective damping factor of the seismic base-isolation system, expressed as below:

$$\beta_{eff} = \frac{4Q(D - D_y)}{2\pi K_{eff} D^2} \tag{7}$$

The LRB is characterized by the isolation period $T_b$ and the normalized yield strengths $F^x{}_y/W = F^y{}_x/W$. Here, $W$ ($m^*g$) is the total weight of the structure; and $g$ is acceleration due to gravity [51,60,61]. The stiffness and damping of the LRB were selected to provide specified values of the two parameters, namely, the isolation period ($T_b$) and damping ratio ($\xi_b$), defined as follows [60,62].

$$T_b = 2\pi \sqrt{\frac{m}{K_{xb}}} \tag{8}$$

$$\xi_b = \frac{\Sigma_j c_{xbj}}{2m\omega_x} \tag{9}$$

Here, $m$ is the total mass of the structure resting on the isolators.

The values of the LRB parameters (pre-yield stiffness ($K_1$), post-yield stiffness ($K_2$)) was taken from similar research after searching many technical documents and research studies. The values of the characteristic parameters and the stiffness ratios considered for the isolation systems are listed in Table 5. The isolation periods of most base-isolated

buildings are within the range between 2.0 s and 3.0 s. However, in parallel to the increase in sizes and capacities of isolation system elements, the fundamental periods of seismically isolated buildings have also increased [63].

**Table 5.** Properties of the lead rubber bearings.

| Type | Linear Stiffness (kN/m) | Yield Strength (kN) | Post Yield/Stiffness Ratio α | Vertical Stiffness (kN/m) | R (m) | d (m) | D (m²) |
|---|---|---|---|---|---|---|---|
| LRB (500 kN) | 322 | 31.0 | 0.1 | 150.720 | 0.400 | 0.354 | 0.0028 |
| LRB (1000 kN) | 644 | 55 | 0.1 | 294.450 | 0.500 | 0.280 | 0.0050 |

Nlprop is a group of structural properties used to describe the behavior of Nllink elements. Each Nlprop consists of six internal nonlinear springs. The force–deformation relationship of these springs can be combined or independent from each other. Each Nlprop specifies a nonlinear force–strain relationship for six internal deformations. These nonlinear properties are used only in nonlinear time history analysis [31].

The form of the link elements was modeled using modelling calculation based on studies by Gazi and Alhan (2019) and Soyluk (2010) [31,63].

A historical masonry mosque with a single dome was considered for this study. Because single-domed and similar mosques are widely found all over the country, the values obtained from the study can form a guide for similar historical mosques. For the process, hard beams (reinforced concrete elements) were placed along the base of the walls to facilitate the application of the base isolation. This configuration allows the insulator to be located between the base and the superstructure. The damped nonlinear spring model was used to implement the FEM model. The application of lead rubber isolation was considered in this study. The number of insulators in the historical mosque was 31 with equal steps; there were 7 isolators with 500 kN vertical loads and 24 isolators with 1000 kN vertical loads at the base of the mosque. Representations of the historical mosque and the isolator details are shown in Figure 4.

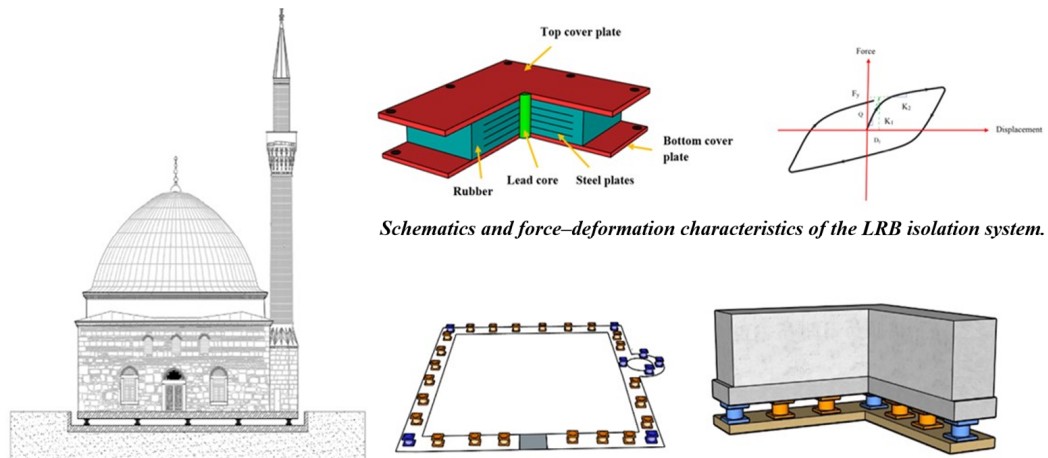

*Schematics and force–deformation characteristics of the LRB isolation system.*

**Figure 4.** Representations of the historical mosque and the isolator details.

## 5. Assessment of Nonlinear Time History Analysis of the Historic Masonry Structure

Firstly, modal analysis was performed on the 3D FEM models to identify the main frequencies, the related modal shapes, and the effective modal masses (% Meff) of each mode of the mosque as a percentage of the total. Three hundred modes were considered in the modal analysis, and the total mass participation rates were found to be 86% for the fixed-base structure and 98% for the base isolation. Generally, for this model, the distribution of the modal shapes clearly shows how the mosque deforms. The second mode had the

highest participating mass in the longitudinal direction (Meff = 98 %, T = 2.28 s) under base isolation. The 259th mode had the highest participating mass in the longitudinal direction (Meff = 86 %, T = 0.015 s) for the fixed-base structure. The distribution and stress of modal shapes in the longitudinal and transversal directions for the models with (a) a fixed-base structure and (b) base isolation is shown in Figure 5. The higher modal shapes of the mosque are a combination of transversal vibration modes and torsional modes. The distribution of the modal shapes demonstrates that the mosque, though characterized by stiff structural elements on the perimeter, displays low transversal and torsional stiffnesses, with significant out-of-plane deformations of the elements. Furthermore, the deformed plan configuration confirms that the seismic loads acting along either the longitudinal or transversal direction involve remarkable out-of-plane deformations of the orthogonal structural elements.

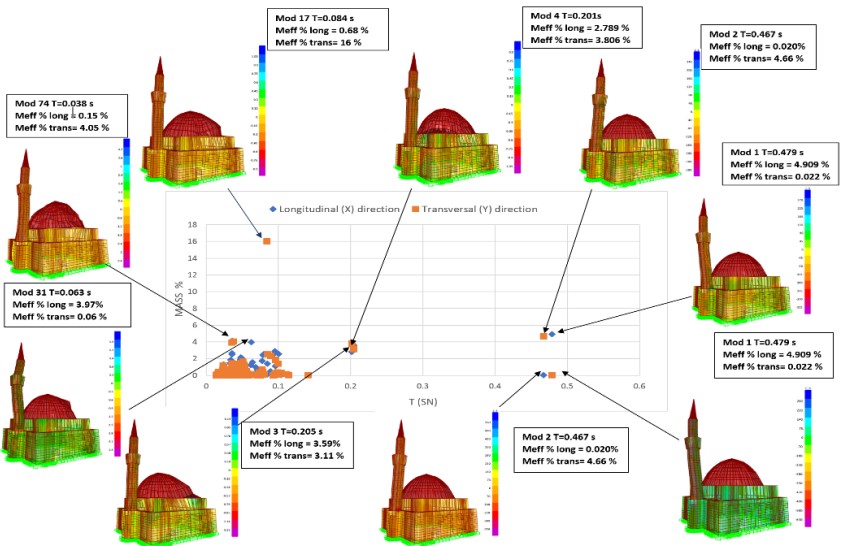

(**a**) The model with a fixed-base structure.

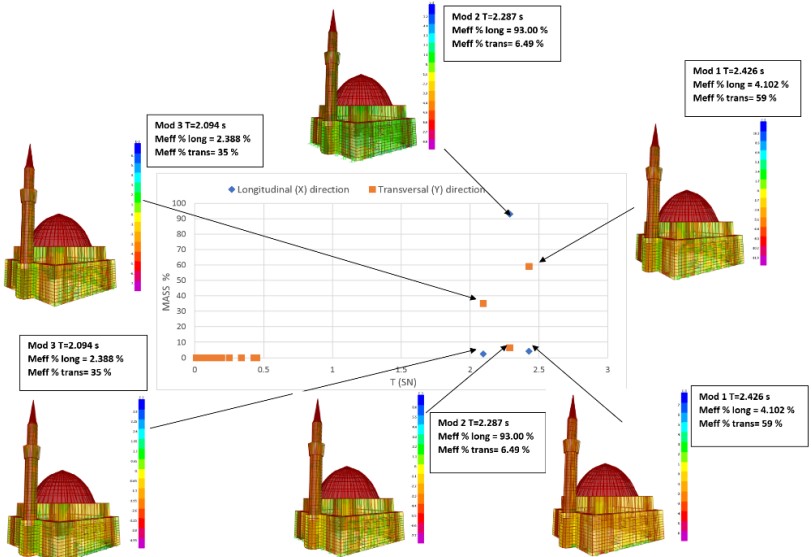

(**b**) The model with base isolation.

**Figure 5.** Distribution and stress of modal shapes in the longitudinal and transversal directions for the models with (**a**) a fixed-base structure and (**b**) base isolation.

## 6. Verification of the High-Damping LRB Isolators

The hysteretic cycles of two representative high-damping LRBs are illustrated in the following. The analysed devices are the high-damping LRB with 500 kN load, located at the four corners of the mosque and the high-damping LRB with 1000 kN load, located at the base of the column. The hysteretic cycles of the high-damping LRBs with 500 kN and 1000 kN vertical loads are shown in Figure 6.

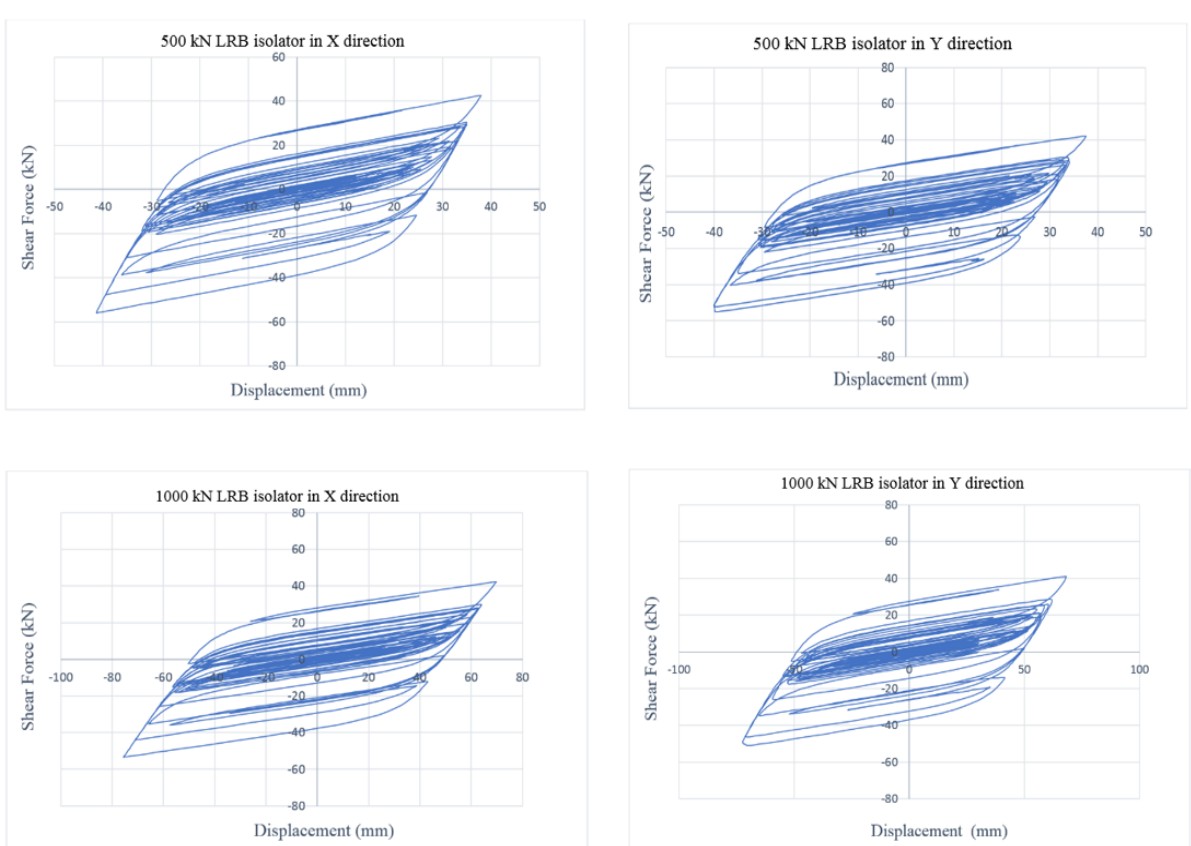

**Figure 6.** The hysteretic cycles of LRB isolators with 500 and 1000 kN vertical loads for seismic actions in the X and Y directions.

## 7. Results

After the application of artificial earthquake load in nonlinear time history analysis, it is possible to appreciate the structural behavior of the historical masonry mosque with or without isolation. The seismic performance of variants was analyzed for the historical masonry mosque with or without base isolation. In both cases, a nonlinear time history analysis was carried out using an artificial earthquake load, generated according to the TBEC (2018) regulation. The first mod value of the historical mosque with a fixed base was 0.478 s. The targeted period value of the systems with a base isolator was chosen to be between 2 and 2.5 and was found to be 2.42 after analysis. Besides this, when the values were examined, it was seen that there were significant increases in period values compared to the built-in system when using seismic isolators for other modes. The modes and periods for the historical mosque are shown in Figure 7.

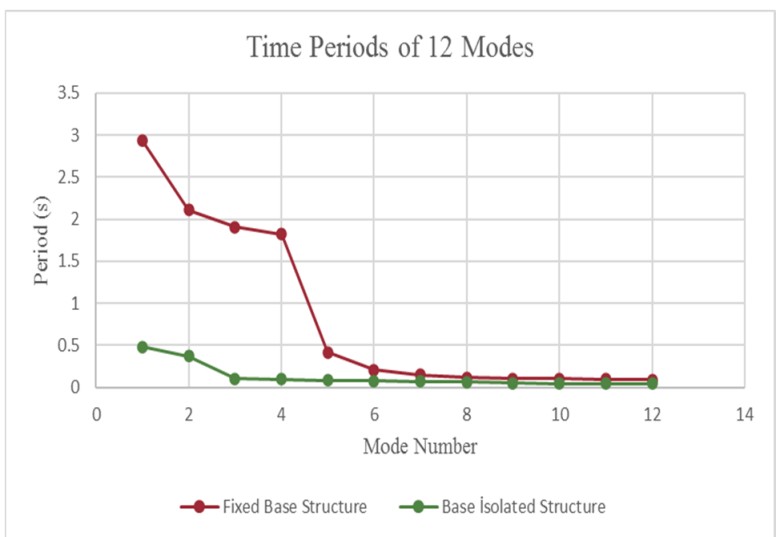

**Figure 7.** The periods and modes of the historical mosque.

The period of the building with a base-isolation structure also increased by 6 times to 2.94 s, as compared to that of the cross-braced structure, which was 0.48 s.

The displacements at each determined level in both the X and Y directions are shown in Figures 8 and 9. It can be observed that the displacements in the base-isolated structure were greater than those in the original structure. This is because there was a large displacement at the foundation level in the base-isolated structure, while the upper levels acted almost as rigid bodies compared to the foundation. However, it may be misleading to compare the determined displacements to decide which case is better. The maximum displacements of the structure in the X direction in the fixed-base analysis at the upper level of the minaret and mosque were $d_{x-max}$ = 5.88 cm and $d_{x-max}$ = 1.55 cm in the transversal direction, respectively. The maximum displacements of the structure in the Y direction in the fixed-base analysis at the upper level of the minaret and mosque were dy-max = 6.01 cm and dy-max = 1.8 cm in the transversal direction, respectively.

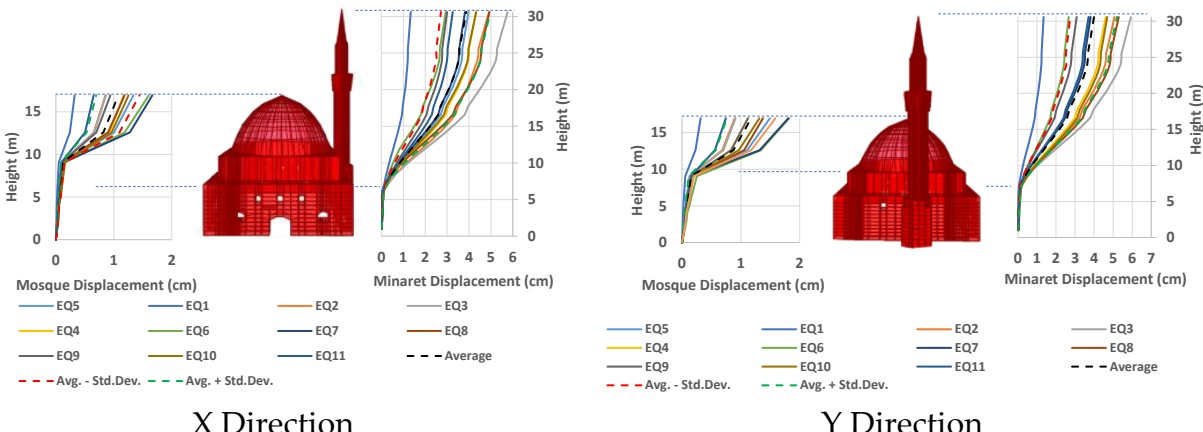

**Figure 8.** Mosque and minaret displacements for the fixed-base case.

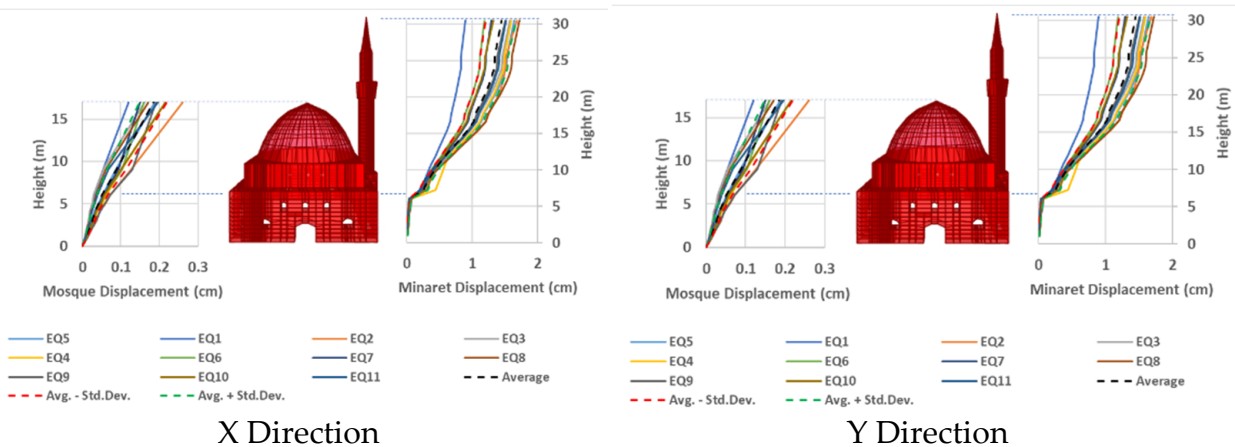

**Figure 9.** Mosque and minaret displacements for the base-isolated case.

The maximum displacements of the structure in the X direction in the base-isolation analysis at the upper level of the minaret and mosque were $d_{x\text{-}max}$ =1.55 cm and $d_{x\text{-}max}$ = 0.25 cm in the transversal direction, respectively. The maximum displacements of the structure in the Y direction in the base-isolation analysis at the upper level of the minaret and mosque were $d_{y\text{-}max}$ = 2.8 cm and $d_{y\text{-}max}$ = 0.17 cm in the transversal direction, respectively.

The displacement values for the fixed-base structure and base-isolated structure configurations are the relative structural displacements, calculated as the difference between the determined points (shown in the figures) for each earthquake. It was observed that for the fixed-base structure, the story displacements were larger compared to those for the base-isolated structure.

The figures of the historical mosque with a fixed base in both the X and Y directions show that the highest absolute acceleration values for minaret levels occurred in the $EQ_3$ earthquake, while the lowest values occurred in the $EQ_1$ earthquake. The highest absolute acceleration values for mosque levels occurred in the $EQ_7$ earthquake, while the lowest values occurred in the $EQ_1$ earthquake. The figures of the historical mosque with base isolation in the X direction show that the highest absolute acceleration values for minaret levels occurred in the $EQ_8$ earthquake, while the lowest values occurred in the $EQ_1$ earthquake. The highest absolute acceleration values for mosque levels occurred in the $EQ_2$ earthquake, while the lowest values occurred in the $EQ_1$ earthquake. The figures of the historical mosque with base isolation in the Y direction show that the highest absolute acceleration values for minaret and mosque levels occurred in the $EQ_4$ earthquake, while the lowest values occurred in the $EQ_1$ earthquake.

Due to the probability of earthquake effects being low, inelastic deformation and displacements (controlled damage that will not cause total collapse) are allowed for structures under the effect of earthquakes. The drift ratio control, which represents the overall behavior of the structure, could be more determinant for structures. Considering the displacement values formed in terms of the elements, the maximum displacement value occurs in the upper part of the minaret and on the façade. Owing to the boundary conditions regarding the horizontal displacement values, it can be seen that the conditions for three different stages are defined in the Regulation on the structures to be built in Earthquake Zones: the minimum damage limit, the safety limit, and the collapse limit. The relative floor displacement values on any floor of the building are foreseen as 0.01 for the minimum damage limit, 0.03 for the safety limit, and 0.04 for the collapse limit (Aslay and Reader, 2020). The maximum relative floor displacement ratio obtained in this study was calculated to be approximately 0.0183 for the minaret peak in both directions and approximately 0.005 for the dome. The calculated value for the minaret is above the minimum damage limit value stated above. This situation suggests that cracks formed and damage occurred as

a result of the earthquake at the upper points of the minaret. Maximum displacements occurred at the top of the minaret.

Figures 10 and 11 show maximum absolute accelerations, which were evaluated by time history analyses, affecting mosque levels and minaret levels. Maximum absolute accelerations are indicators of inertial forces acting at the determined levels due to ground motions, which means that the increase in absolute acceleration at determined levels will increase the earthquake force acting on that determined level. From this perspective, the reduction of absolute acceleration acting at the determined levels by the use of isolators is important progress. According to the analyses made, absolute acceleration decreased significantly in both the X and Y directions due to the application of isolators in the historical mosque.

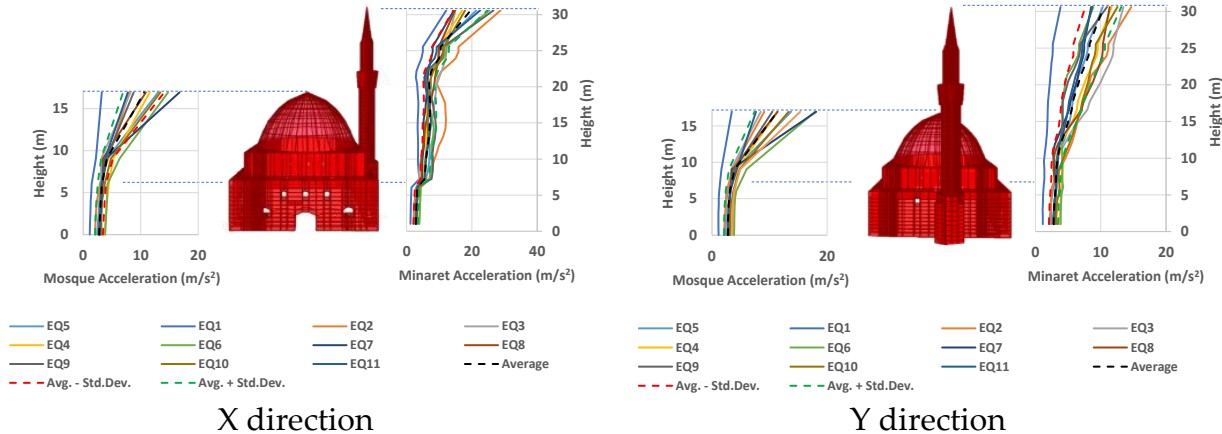

**Figure 10.** Mosque and minaret absolute acceleration for the fixed-base case.

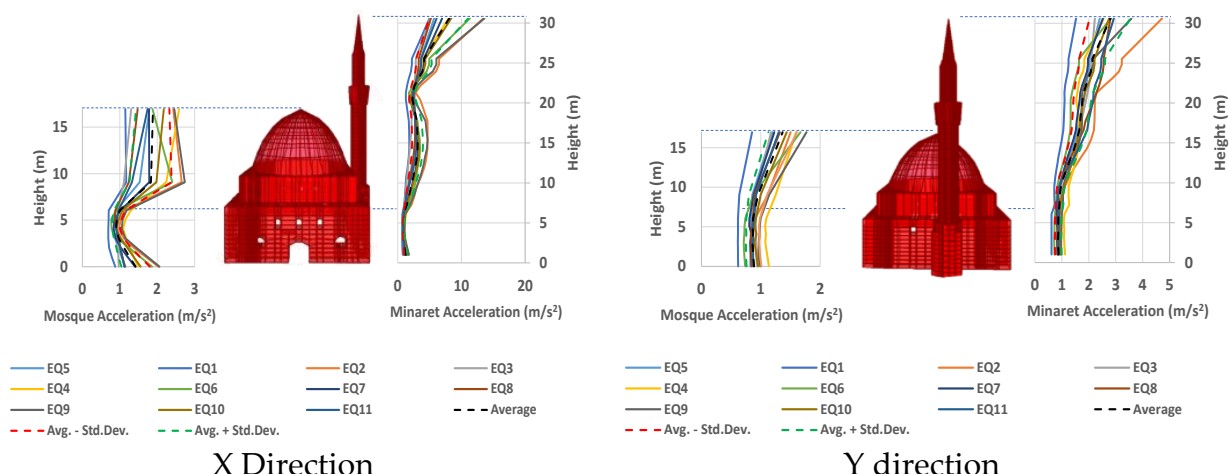

**Figure 11.** Mosque and minaret absolute acceleration for the base-isolation case.

Figures 10 and 11 show the maximum absolute accelerations at the determined levels, which were evaluated by time history analyses. It can be observed that the increase in stiffness compared to the LRB isolator caused an increase in acceleration values, even though it caused a decrease in displacements.

Figures 10 and 11 show that average mosque level acceleration during earthquakes in the base-isolated building model decreased by 83% and by 92% for the X-direction and Y-direction, respectively, and average minaret level acceleration during earthquakes in the base-isolated building model decreased by 63% and by 82% for the X-direction and Y-direction, respectively.

There was a large difference in story acceleration for the fixed-base building model from the bottom to the top. In the base-isolated model, the story accelerations were nearly the same from the bottom to the top. The highest absolute acceleration values for minaret levels occurred in the $EQ_2$ earthquake, while the lowest values occurred in the $EQ_1$ earthquake. The highest absolute acceleration values for mosque levels occurred in the $EQ_9$ earthquake, while the lowest values occurred in the $EQ_7$ earthquake.

## 8. Conclusions

The main purpose of this paper was to discuss the application of a base-isolation system using the finite element technique for seismic assessment of historic masonry buildings, focusing on how the base-insulator system affects historical masonry behavior under earthquake load. For the base-isolation system, two types of LRB isolators with different load capacities were placed. One of the reasons for using a mixed base-isolation system consisting of two different load-capable insulators is to reduce the total cost of the isolation system.

The results obtained from the analysis showed that the use of base insulation significantly reduced the occurrence of earthquake-induced negative effects in the historical building. Some damage occurred in the minarets and bodies of historical buildings when the values emerging during earthquakes exceeded the strength values of the building. On the other hand, the maximum strength values occurring in the base-insulated structure were below the limit value compared to those in the built-in support structures. The results obtained show that the application of rubber base insulators can prevent the destructive effects of earthquakes. In this context, when the magnitude of the earthquakes applied to our building is taken into consideration, it is concluded that if the studied historical building is strengthened using an isolator, it will preserve its integrity in earthquakes of magnitude up to 6.5. This situation will contribute to the protection and future transfer of cultural heritage locally and nationally. Finally, the results can be useful in assessing the existing earthquake resistance of historic buildings in order to strengthen similar existing historic buildings and/or prepare emergency plans. The vulnerability of other cultural heritage found in seismic areas can thus be minimized.

The results of this paper indicate that base isolation is quite effective in reducing the displacement of historical masonry structures. Thus, we can clearly say that base isolation is significantly effective in the seismic retrofitting of existing historical masonry structures. Protection systems involving base isolation are usually installed at the foundation or underground level, so minimum intervention is required in the installation of a foundation insulation layer, thus preserving the historical texture of the structures.

Base isolation security improvements will considerably improve disaster management for such historical buildings during earthquakes and reduce post-earthquake repair costs.

**Funding:** This research received no external funding.

**Institutional Review Board Statement:** Not applicable.

**Informed Consent Statement:** Not applicable.

**Data Availability Statement:** Not applicable.

**Conflicts of Interest:** The author declares no conflict of interest.

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
