# Peer review of "Investigation of a Base-Isolator System’s Effects on the Seismic Behavior of a Historical Structure"

_buildings, doi:10.3390/buildings11050217_

Round 1

Reviewer 1 Report

The paper presents a study about the seismic behavior and performance of a mosque with and without the use of base isolation system. The topic is interesting and it gains traction among researchers. Nevertheless, before to consider the paper suitable for publication, it needs of some improvements and clarifications, as well as a review of the English. The authors can see this reviewer's comments in the attached PDF.

Author Response

Comments and Suggestions for Authors

The paper presents a study about the seismic behavior and performance of a mosque with and without the use of base isolation system. The topic is interesting and it gains traction among researchers. Nevertheless, before to consider the paper suitable for publication, it needs of some improvements and clarifications, as well as a review of the English. The authors can see this reviewer's comments in the attached PDF.

  • Corrections requested by you and the other referee have been marked on the main text.
  • First of all, thank you for your interest.
  • We reviewed and reshaped the our study according to you suggested.
  • All are available corrected in the appendix. You can see the changes related to the correction pop-up note.

Kind Regards,

Strong Healthy days.

Reviewer 2 Report

It is interesting that this manuscript presents the nonlinear time history analyses for historical masonry structure and analysis the base-isolated system on  seismic behavior of historical masonry structure. 
There are some points where the paper should be improved.

1. The manuscript has no literature review about the base isolation system on the seismic behavior of historical structure or other structure. It would be 
very important to see the previous research or at least a small summary about the previous research activities in this topic.

2. What do the two pictures show in Figure.2 ? It should add the name of the two pictures and explained them clearly.

3. "The earthquake records were scaled to ensure that the spectra in question provided the desired properties." What is the principle or method of the 
earthquake records scale?

4. The elastic stiffness should equal the force divided by the displacement, Eq.(4) "-" should modify to "/". And how to get Eq.(2)? These equations are 
descried confusing and should be well statement.

5. What are the differences among Figures. 12-15? Please explain it clearly.

6.Some of the research addressing these issues should be acknowledged, some recommended references, among many others are: 
DOI: 10.1016/j.engstruct.2019.109553;DOI: 10.3311/PPci.15276;DOI: 10.1016/j.conbuildmat.2011.11.002;DOI: 10.1016/j.istruc.2020.12.089.

Author Response

Comments and Suggestions for Authors

It is interesting that this manuscript presents the nonlinear time history analyses for historical masonry structure and analysis the base-isolated system on  seismic behavior of historical masonry structure. 
There are some points where the paper should be improved.

  1. The manuscript has no literature review about the base isolation system on the seismic behavior of historical structure or other structure. It would be  very important to see the previous research or at least a small summary about the previous research activities in this topic.

For 1 command;

Necessary addition has been made

  1. What do the two pictures show in Figure.2 ? It should add the name of the two pictures and explained them clearly.

For 2 command;

One of them is finite element fixed model, the other is finite element model of based isolation.

All show in text .

  1. "The earthquake records were scaled to ensure that the spectra in question provided the desired properties." What is the principle or method of the  earthquake records scale?

For 3 command;

5% damped SRSS (scaled) response spectra of all 11 ground motion records were scaled to the target design spectra assumed in this study (DD2 TBDY).

Total 11 earthquake data records are selected to conduct Time history analysis. In the analyzes, a total of three acceleration components, two horizontal and one vertical component, were used for each earthquake. Earthquake records were scaled to fit target earthquake spectrum.

  1. The elastic stiffness should equal the force divided by the displacement, Eq.(4) "-" should modify to "/". And how to get Eq.(2)? These equations are  descried confusing and should be well statement.

For 4 command;

Re-edit these equations

  1. What are the differences among Figures. 12-15? Please explain it clearly.

For 5 command;

Re-edit these figures

6.Some of the research addressing these issues should be acknowledged, some recommended references, among many others are: 

DOI: 10.1016/j.engstruct.2019.109553;

DOI: 10.3311/PPci.15276;

DOI: 10.1016/j.conbuildmat.2011.11.002;

DOI: 10.1016/j.istruc.2020.12.089.

Corrections requested by you and the other referee have been marked on the main text.

First of all, thank you for your interest.

We reviewed and reshaped the our study according to you suggested.

All are available corrected in the appendix. You can see the changes related to the correction pop-up note.

Kind Regards,

Strong Healthy days.

Reviewer 3 Report

The topic is interesting but the paper, in the present format, requires further corrections in order to become suitable for publication.

1- The abstract can be improved.

2- Similarly, the introduction do not introduces a significant literature review.

3- Please provide more information about the selected structure.

Author Response

Dear to whom (review3)

The requests of another review would be similar to your requests for correction.
We also made changes in that direction. I share with you the marked changes. Hope it is enough for you.

We look forward to your referrals. According to it, I believe we will improve our publication to a higher level.

This is an uploaded paper doc in below.

Sincerely

Strong Healthy Days,

Authors (BP)

Round 2

Reviewer 1 Report

The revised version of the paper presents some modifications and improvements but there is not the tracking of these and, above all, there is not a report containing the answers to the questions made by this Reviewer. 

Before a final acceptance, I would like to see this file. 

Author Response

Dear Referee
I redirect the pop-up file that we have made the corrections you said in the attachment.

First of all, thank you for making such a detailed review of my studies.
I added the publications you suggested to my studies. I have added similar publications in the local area. I read/looked up the publications, you said.
While there are normally around 50 references, now there are 60 references.
You said to fix the spelling mistakes, fixed them all.
I wrote where I got the formulas and added them again by typing math type.
I detailed how I used how to connect the isolator in modelling.
In particular, I took the thesis made in Architecture at Gazi University as a reference. In this PhD thesis, there were different modelling of buildings and historical artefact modelling.
All calculations are calculated according to the formulas in this thesis.
I have further developed the knowledge of the building.
I developed and explained the modelling a little more.

I hope it has been a suitable study for your directions.

Kind regards,

Author

Reviewer 3 Report

The draft is suitable for publication, provided that some minor spell and typographic errors are discussed with the editing team (see for example the doted capital I in the title, in the word: İsolator).

Round 3

Reviewer 1 Report

The paper can be accepted for publication